# A novel mechanism of gland formation in zebrafish involving transdifferentiation of renal epithelial cells and live cell extrusion

Richard W Naylor, Hao-Han G Chang, Sarah Qubisi, Alan J Davidson*

Department of Molecular Medicine and Pathology, University of Auckland, Auckland, New Zealand

**Abstract** Transdifferentiation is the poorly understood phenomenon whereby a terminally differentiated cell acquires a completely new identity. Here, we describe a rare example of a naturally occurring transdifferentiation event in zebrafish in which kidney distal tubule epithelial cells are converted into an endocrine gland known as the Corpuscles of Stannius (CS). We find that this process requires Notch signalling and is associated with the cytoplasmic sequestration of the Hnf1b transcription factor, a master-regulator of renal tubule fate. A deficiency in the Irx3b transcription factor results in ectopic transdifferentiation of distal tubule cells to a CS identity but in a Notch-dependent fashion. Using live-cell imaging we show that CS cells undergo apical constriction *en masse* and are then extruded from the tubule to form a distinct organ. This system provides a valuable new model to understand the molecular and morphological basis of transdifferentiation and will advance efforts to exploit this rare phenomenon therapeutically.
DOI: https://doi.org/10.7554/eLife.38911.001

## Introduction

Transdifferentiation is the poorly understood process by which a mature cell is converted to a completely different cell type (*Tosh and Slack, 2002*). Under normal biological settings, transdifferentiation is a rare phenomenon, although it can be readily induced experimentally by forcing the expression of genes that direct cell fate changes, such as the classic example of the MyoD transcription factor being able to convert fibroblasts into muscle lineage cells (*Tapscott et al., 1988*). The process of transdifferentiation can be either direct, whereby a mature cell converts seamlessly into another mature cell type, or indirect, whereby there is a requirement for dedifferentiation to a more immature intermediary cell type, followed by differentiation into the new fate (*Thowfeequ et al., 2007*; *Merrell and Stanger, 2016*). Examples of transdifferentiation occurring in a developmental setting are uncommon, the best studied cases are in *C. elegans* embryos with the indirect transdifferentiation of rectal epithelial Y cells into cholinergic motor neurons (*Jarriault et al., 2008*) and the formation of MCM interneurons from AMso glial cells (*Sammut et al., 2015*).

In vertebrates, direct transdifferentiation is largely limited to the adult setting where it is associated with response to injury. For example, ablation of pancreatic β-cells induces the transdifferentiation of resident α-cells to β-cells in both mice and zebrafish (*Thorel et al., 2010*; *Ye et al., 2015*). Similarly, in the liver, chronic injury promotes the conversion of hepatocytes to biliary epithelial cells through the combined action of the Notch and Hippo signalling pathways (*Yanger et al., 2013*). Cases of indirect transdifferentiation in vertebrates include the well-known example of lens regeneration in amphibians following lentectomy (*Stone, 1967*), in which retinal pigmented epithelial cells initiate expression of pluripotency genes (*Maki et al., 2009*), dedifferentiate and then mature into lens cells (*Sánchez Alvarado and Tsonis, 2006*). Indirect transdifferentiation is considered to occur in some cancers, via the epithelial-to-mesenchymal transition and dedifferentiation that often

*For correspondence:
a.davidson@auckland.ac.nz

**Competing interests:** The authors declare that no competing interests exist.

accompanies tumourigenesis (*Shekhani et al., 2013*; *Maddodi and Setaluri, 2010*; *Maniotis et al., 1999*; *Fang et al., 2005*). In summary, while transdifferentiation in vivo is possible under normal and pathogenic settings, it remains a rare and poorly understood phenomenon.

The zebrafish offers a visually accessible vertebrate model with which to study cell fate changes in the context of organogenesis. The embryonic kidney (pronephros) is particularly well-suited for these studies because of its readily visualised location within the embryo and a high degree of understanding of how cell division, differentiation and morphogenesis are co-ordinated during organ formation (*Drummond et al., 1998*; *Majumdar et al., 2000*; *Wingert and Davidson, 2011*; *Wingert et al., 2007*; *Wingert and Davidson, 2008*; *Naylor et al., 2013*; *Naylor et al., 2016b*; *Naylor et al., 2017*). The zebrafish pronephros is analogous to the filtering units in the mammalian kidney (nephrons) and consists of a midline-fused blood filter (glomerulus), attached to bilateral renal tubules that extend to the cloaca (*Drummond et al., 1998*; *Wingert et al., 2007*; *Wingert and Davidson, 2008*; *Drummond and Davidson, 2010*). The tubules are subdivided into functionally distinct segments consisting of the proximal convoluted tubule (PCT), the proximal straight tubule (PST), the distal early tubule (DE), and the distal late segment (DL; Figure 1 and [*Wingert et al., 2007*]). Each tubule segment expresses a specific set of genes that defines its functional differentiation. The PCT and PST are associated with bulk re-absorption of solutes from the filtrate and express a wide variety of solute transporters (*Wingert et al., 2007*; *Blaine et al., 2015*; *Ullrich and Murer, 1982*). In contrast, the DE and DL segments express fewer transporters, suggesting that they function more to fine-tune the composition of the filtrate. For example, functionality of the DE segment is conferred by the expression of *slc12a1*, encoding a Na-K-Cl co-transporter (*Wingert et al., 2007*). The transcription factor Hnf1b is a major regulator of renal tubule identity in both zebrafish and mammals (*Naylor et al., 2013*; *Heliot et al., 2013*; *Massa et al., 2013*). In *hnf1b*-deficient zebrafish embryos, solute transporter expression fails to initiate in the tubule segments and the kidney tubules remain as a simple, non-transporting, tubular epithelium (*Naylor et al., 2013*).

Above the pronephric tubules, and initially forming at the junction between the DE and DL segments, is positioned an endocrine gland called the Corpuscles of Stannius (CS, *Figure 1*). The CS gland in teleost fish is responsible for secreting Stanniocalcin-1 (Stc1), a hormone involved in the homeostatic regulation of calcium (*Cheng and Wingert, 2015*; *Tseng et al., 2009*). The CS glands consist of lobes of epithelial cells that are separated by strands of connective tissue containing blood vessels and nerve tracts, which is quite distinct from the cuboidal polarised epithelium that constitutes the pronephric tubule (*Menke et al., 2011*; *Cohen et al., 1975*). As such, the CS gland is a functionally and structurally distinct organ that shares little homology, other than an epithelial state, to the kidney. Despite this, early observations have led to the suggestion that the CS gland originates from the pronephric tubules, although its formation has not been extensively characterised.

In this study, we investigate the origin of the CS gland and discover that it arises by a previously uncharacterised process of direct transdifferentiation from DE tubule cells followed by apical constriction and extrusion. We find that the molecular pathways that control this transdifferentiation event involve Notch signalling, cytoplasmic sequestration of the renal identity factor Hnf1b, and inhibitory signals conferred by the Iroquois transcription factor Irx3b. Together, these results demonstrate a rare example of direct transdifferentiation under normal physiological conditions in a vertebrate.

## Results

### CS cells originate in the renal tubule and are extruded to form a gland

To better understand CS gland formation, we first examined expression of *stanniocalcin-1* (*stc1*), a marker of CS fate, and *cdh17*, a pan-renal tubule marker that encodes a cadherin involved in cell-to-cell adhesion (*Horsfield et al., 2002*). We found that at 24 hr post-fertilisation (hpf), transcripts for *cdh17* are down-regulated in the posterior most portion of the DE segment (*Figure 1A*). Concomitant with this, the first $stc1^+$ cells appear in this region and increase in number. By 32 hpf the $stc1^+$ appear as a prominent bulge on the dorsal side of the tubule and at 50 hpf, they are found as discrete, and separate, structures on top of each pronephric tubule in ~70% of animals (*Figure 1C*). These observations suggest that CS cells arise from renal epithelial cells via transdifferentiation and are then physically extruded from the tubule.

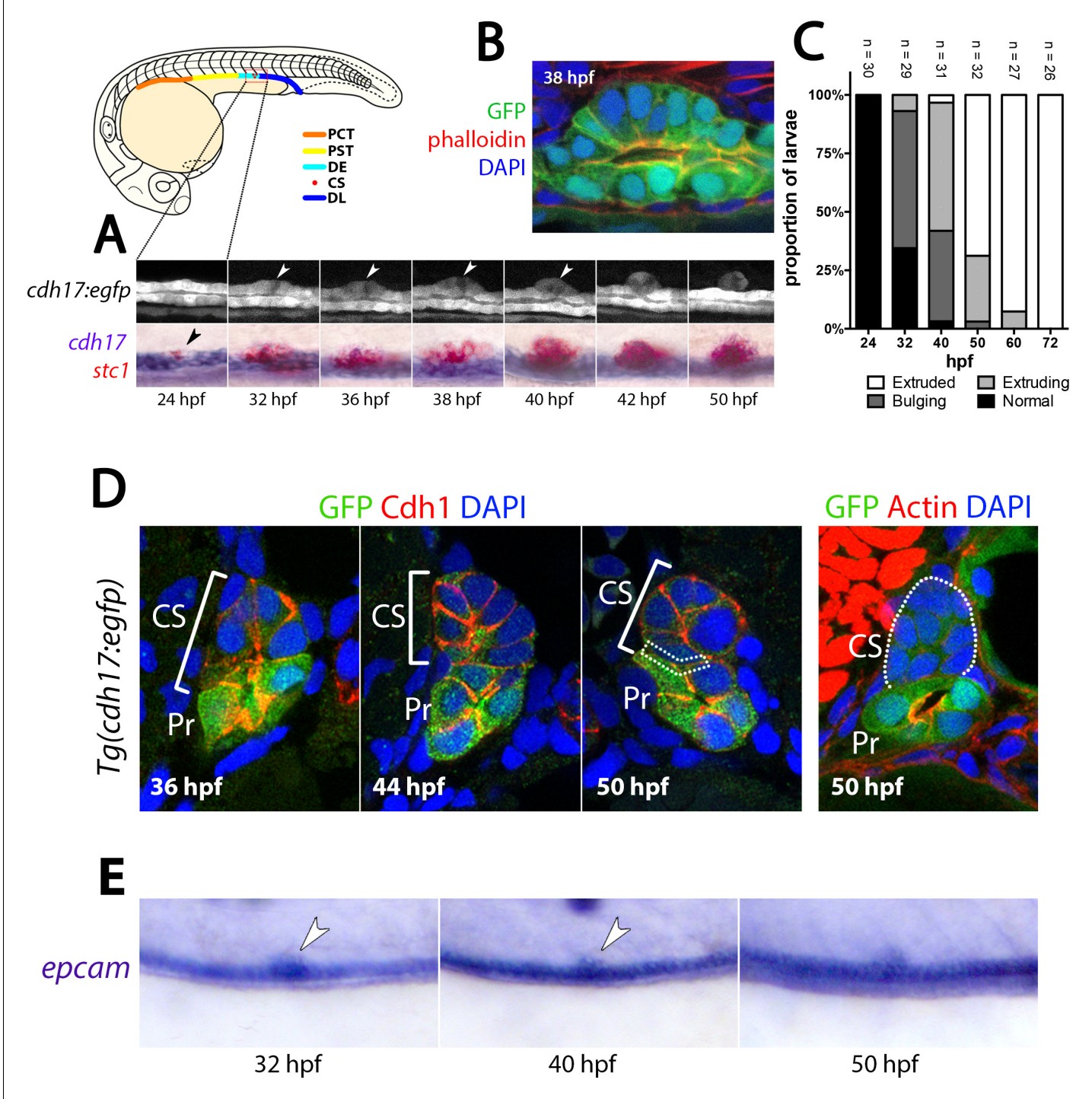

**Figure 1.** CS extrusion is achieved by apical extrusion of multiple renal epithelial cells. (**A**) Lateral views of a stage series analysis of CS extrusion in a single live *Tg(cdh17:egfp)* embryo (top panels) and embryos fixed at the stages shown and stained for *cdh17*(purple)/*stc1*(red). (**B**) Image of a sagittal section through the pronephros of a *Tg(cdh17:egfp)* embryo co-labelled with Phalloidin (F-actin, red) and DAPI (nuclear stain, blue) at the site of the extruding CS at 38 hpf. (**C**) Histogram shows the frequency of the four stages of CS extrusion at 24 hpf, 32 hpf, 40 hpf and 50 hpf. (**D**) Panels show transverse sections through the CS gland of *Tg(cdh17:egfp)* embryos at the stages indicated. Green fluorescence is from the endogenous GFP, Cdh1 is labelled red and nuclei are labelled blue (DAPI). Dotted box in the 50 hpf panel indicates weak/absent Cdh1 staining at the interface between the ventral side of the CS gland and the dorsal side of the tubule. (**E**) Panels show lateral views of an extruding CS gland in embryos at the indicated stages labelled with *epcam*. Abbreviations: PT, Proximal tubule; DE, Distal early tubule segment; CS, Corpuscles of Stannius; DL, Distal Late tubule segment; hpf, hours post fertilisation; Pr, Pronephros.

DOI: https://doi.org/10.7554/eLife.38911.002

To visualise the process of CS gland formation, live time-lapse imaging was performed on *Tg (cdh17:egfp)* embryos from 24 to 50 hpf (*Figure 1A* and *Video 1*). Presumptive CS cells were observed to bulge out of the dorsal wall of the tubule concomitant with the constriction of their apical membranes. Immunostaining of sagittal cross-sections through the CS region at this stage showed that the cells adopt a wedge shape with narrowed apical membranes (marked by Phalloidin⁺ F-actin, *Figure 1B*). As apical constriction increased, we observed that the adjacent non-CS epithelial cells moved closer together and the CS cells protruded dorsally as an arch two cell layers wide (*Figure 1D*). The epithelial cells on either side of the CS cells eventually met at 42 hpf in most animals, followed by extrusion of the CS gland from the tubule (*Figure 1A* and *Video 1*). The timing of extrusion was variable but was complete in the majority of embryos by 60 hpf (*Figure 1C*). We found that during extrusion from the tubule, the *epithelial cell adhesion molecule* gene (*epcam*) remains continuously expressed in CS cells (*Figure 1E*) and the major epithelial cell adhesion protein Cadherin-1 (Cdh1) is retained on the basolateral membranes (*Figure 1D*), suggesting that the epithelial status of CS cells is not changed during extrusion. After CS cell extrusion, Cdh1 is detectable at the interface between CS and tubular cells and is prominent on the lateral membranes of CS cells, similar to renal epithelial cells (*Figure 1D*). However, once extruded Cdh1 immunostaining is largely lost from the interface between CS cells and tubular cells but remains on the basal surface of some CS cells (*Figure 1D*, dotted box region). In addition, F-actin staining, which demarcates the apical surface of tubular cells, is lost in the extruded CS cells (*Figure 1D*). These results suggest a model in which renal epithelial cells transdifferentiate directly into CS cells, without a loss of epithelial character, extrude from the tubule following apical constriction, and form a distinct ball of unpolarised glandular epithelium.

One possibility is that the renal epithelial cells in the tubular region that gives rise to the CS have retained some progenitor-like state and direct transdifferentiation is not occurring in the strictest sense. To explore this, we closely examined the expression of the renal progenitor marker *lhx1a* from 14 to 50 hpf. At 14 hpf, transcripts for *lhx1a* are found throughout the intermediate mesoderm in mesenchymal cells but are rapidly downregulated in all tubule precursors by 16 hpf (*Figure 2— figure supplement 1*), when the transition to an epithelial tube occurs (*Gerlach and Wingert, 2014*; *Naylor et al., 2016a*). This transition corresponds with the restriction of other markers of renal progenitor state, such as pax2a (*Majumdar et al., 2000*) and pax8 (*Tosh and Slack, 2002*), although expression of these genes are retained in the DL segment where they likely play later roles in growth (*Bouchard, 2004*; *Torres et al., 1995*). To demonstrate that the tubule has achieved an advanced state of epithelial differentiation prior to *stc1* transcripts appearing, we confirmed at 22 hpf that the DE segment contains apically localised F-actin and basally deposited Laminin and Collagen type IV, indicative of apical tight junctions and a basolateral basement membrane, respectively (*Figure 2A*). In addition, the DE segment at this time point uniformly expresses *slc12a1*, *atp1a1a.4* and the *krt18* (but not *stc1*), consistent with having the identity of a differentiated transporting epithelium (*Figure 2A*). At 50 hpf, the DE segment continues to be marked by the expression of *slc12a1*, *atp1a1a.4* and *krt18* whereas the extruded CS cells do not express these genes. Instead, staining with Phalloidin and antibodies to Laminin and Collagen type IV shows that the CS gland has lost apically localised F-actin and is enveloped by a separate basement membrane (*Figure 2A*). Taken together, these results support the notion that by 22 hpf, the DE segment

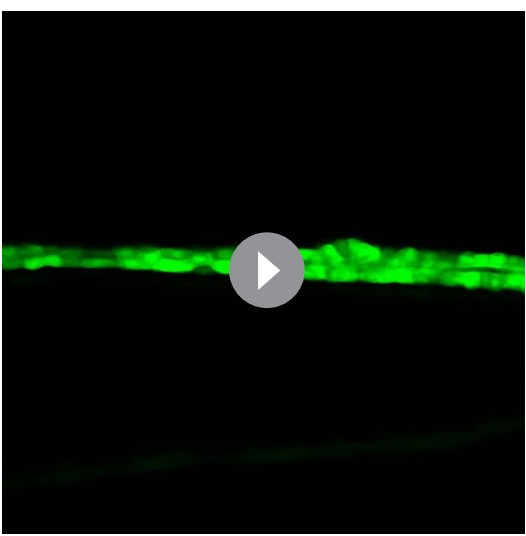

**Video 1.** Time-lapse imaging showing extrusion of the CS endocrine gland from the pronephric tubule Movie shows a lateral view of the posterior trunk region of a Tg(cdh17:egfp) embryo between 32 hpf and 48 hpf. Green fluorescent cells label the pronephric tubule and the CS cells, with the latter observed as a bulge of cells that undergo extrusion during the time-lapse.
DOI: https://doi.org/10.7554/eLife.38911.003

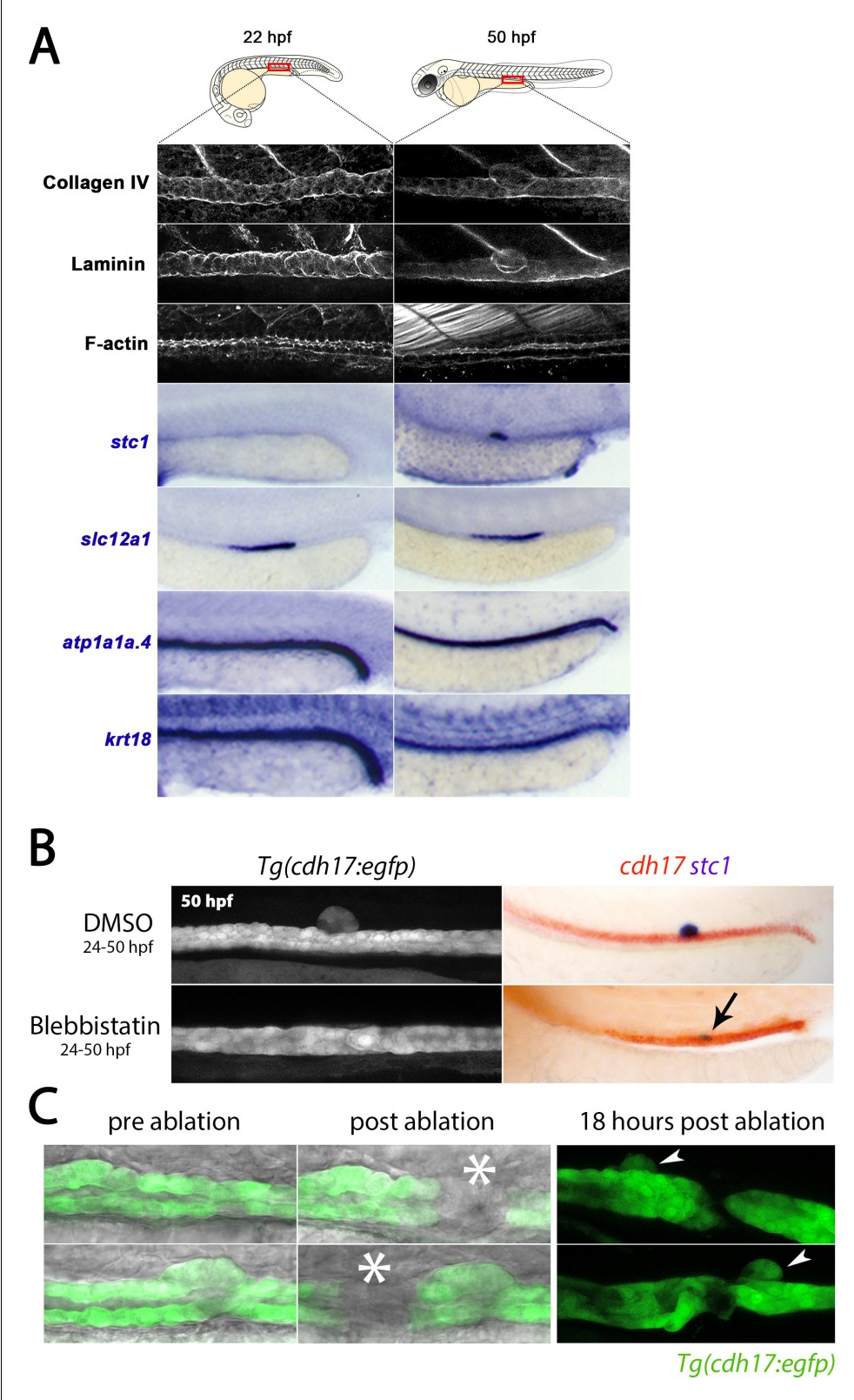

**Figure 2.** CS extrusion is mediated by apical constriction. (**A**) Panels show lateral views of the markers indicated in the region of the trunk where the CS will be derived (at 22 hpf) and is extruded (at 50 hpf). (**B**) Panels show lateral views of the trunk from live *Tg(cdh17:egfp)* embryos (left) and embryos stained for *cdh17* (red)/*stc1* (purple) transcripts (right) in control and Blebbistatin treated embryos. Black arrow indicates the position of *stc1*+ cells in the

*Figure 2 continued on next page*

*Figure 2 continued*

Blebbistatin treated embryo. (C) Panels show brightfield/fluorescence in *Tg(cdh17:egfp)* embryos before and after laser ablation (site of ablation indicated with asterisk). Panels on the right show embryos 18 hr post-ablation and arrowheads indicate the extruded CS gland.

DOI: https://doi.org/10.7554/eLife.38911.004

The following figure supplements are available for figure 2:

**Figure supplement 1.** *lhx1a* expression stage series.

DOI: https://doi.org/10.7554/eLife.38911.005

**Figure supplement 2.** Cell division is not required for CS formation.

DOI: https://doi.org/10.7554/eLife.38911.006

has differentiated into a renal tubular epithelium, and that a subset of these cells then transdifferentiate into CS cells.

## Role of apical constriction, proliferation and neighbouring cells during CS gland extrusion

To determine if CS cell extrusion was dependent upon apical constriction, embryos from 24 hpf onwards were incubated in Blebbistatin (10 µM), an inhibitor of Myosin IIA that blocks the contraction of the apical ring of F-actin filaments, which is essential for apical constriction in other contexts (*Lewis, 1947*; *Sawyer et al., 2010*; *Kovács et al., 2004*). We found that the CS cells failed to extrude in all Blebbistatin-treated embryos (n = 57) and instead, $stc1^+$ cells remained within the tubule (*Figure 2B*). Thus we conclude that apical constriction of CS cells is a necessary prerequisite for extrusion from the renal tubule.

We next examined the role of cell proliferation in CS gland formation by performing EdU incorporation experiments. This analysis showed that CS cells only proliferate once they have been extruded from the tubule (n = 6 at each time-point, *Figure 2—figure supplement 2A*). Inhibition of proliferation between 20 hpf (20-somite stage) and 26 hpf with hydroxyurea and aphidicolin did not reduce the initial number of $stc1^+$ cells that arise in the tubule nor does it block CS cell extrusion (n = 27/27, *Figure 2—figure supplement 2B*). Consistent with CS cells undergoing cell division following their extrusion, fewer $stc1^+$ cells were observed in Blebbistatin-treated embryos at later stages compared to controls, suggesting that there are two phases of CS cell formation: an intra-tubule transdifferentiation phase where a small number of CS cells are generated and a post-extrusion expansion phase.

Previously studied examples of live cell extrusion describe an actomyosin contractile ring that forms in cells neighbouring the extruding cell (*Eisenhoffer et al., 2012*; *Gu et al., 2011*). As all pronephric tubule cells contain apically localised F-actin, it was not possible to determine if an equivalent actomyosin ring forms in the cells surrounding the CS cells. However, laser ablation to sever the tubule (and any interconnected contractile filaments) immediately anterior or posterior to the forming CS gland did not prevent CS extrusion (n = 6, *Figure 2C*), suggesting that neighbouring cells do not play an active role in CS extrusion.

## Nuclear export of Hnf1b occurs in DE cells that transdifferentiate to CS

We next investigated the molecular mechanisms governing the transdifferentiation of DE tubule cells into CS cells. We first examined Hnf1b as this transcription factor is critical for establishing and maintaining renal epithelial cell fate but is not expressed in the extruded CS gland ( (*Naylor et al., 2013*) and *Figure 3—figure supplement 1*). Immunostaining of embryos at 18 hpf showed that all cells in the renal tubule display nuclear localised Hnf1b (n = 7, *Figure 3A*). However, from 22 hpf onwards, we found that Hnf1b is lost from the nucleus in presumptive CS cells and becomes localised to the cytoplasm in a speckled pattern (n = 9, *Figure 3A*, see *Figure 3—figure supplement 2* for lateral view). As this cytoplasmic localisation of Hnf1b precedes the onset of *stc1* expression, it suggests that loss of Hnf1b activity (via cytoplasmic sequestration) may be an early event in the transdifferentiation of renal to CS fate. To provide further support for this notion, we examined *hnf1b*-deficient embryos as we reasoned there might be ectopic CS cell formation in the absence of Hnf1b activity. In line with this, we detected a broad stretch of *stc1* expression in the middle of the pronephric tubule, largely corresponding to the region that would normally form the DE segment (n = 11/11 *hnf1b*-deficient embryos, *Figure 3B*).

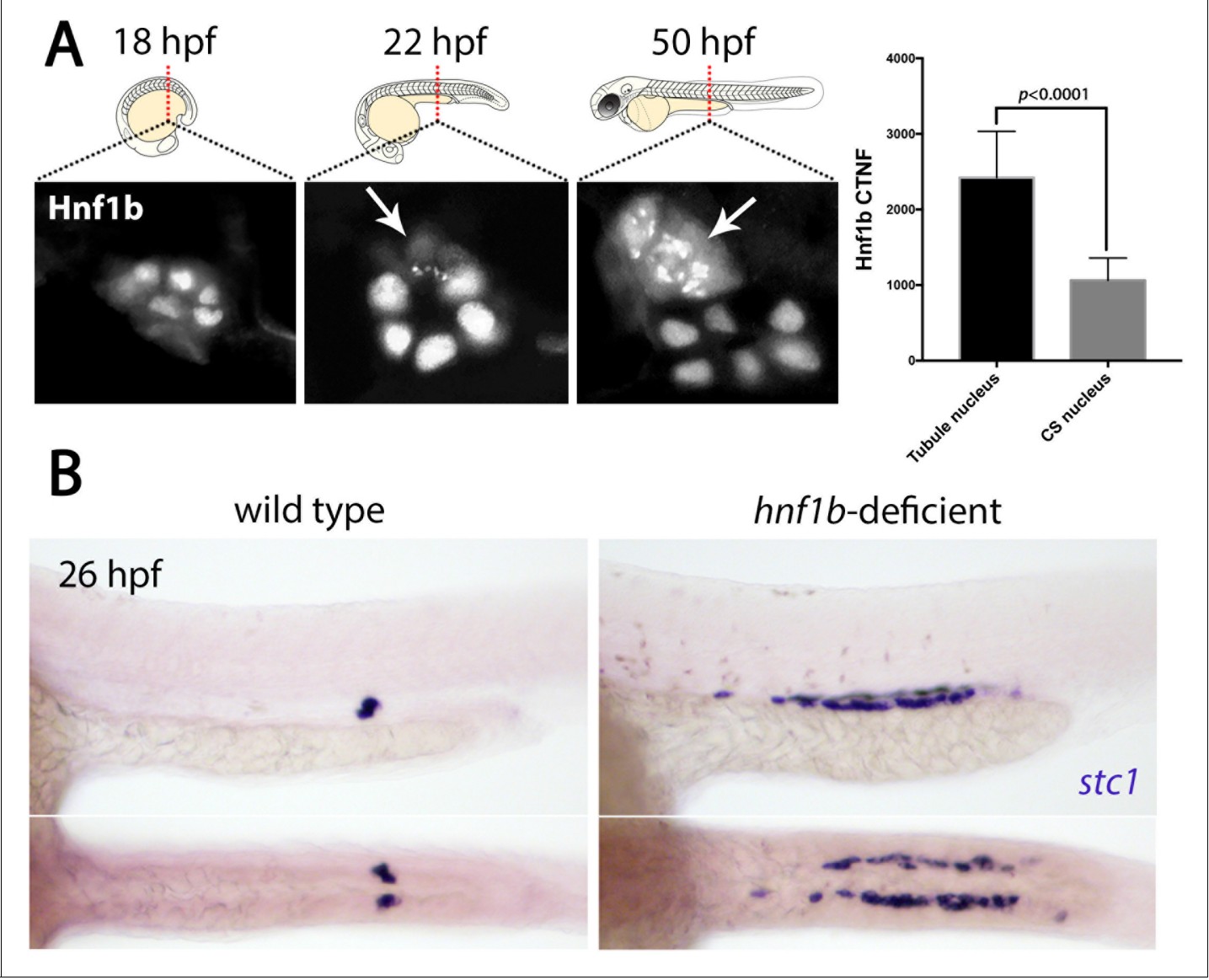

**Figure 3.** Hnf1b is translocated to the cytoplasm in prospective CS cells. (**A**) Panels show Hnf1b immunostains (graysale) on transverse cryosections of zebrafish embryos through the region of the trunk where CS glands form at 22 hpf, 36 hpf and 50 hpf. Arrows indicate CS cells that have undergone nuclear export of Hnf1b. Histogram on the right indicates 'corrected total nuclear fluorescence' (CTNF) of nuclei in the tubule versus nuclei in the CS gland at 22 hpf (n = 8). (**B**) Top panels show lateral views and bottom panels show dorsal views of wild-type and *hnf1b*-deficient embryos stained for *stc1*.

DOI: https://doi.org/10.7554/eLife.38911.007

The following figure supplements are available for figure 3:

**Figure supplement 1.** *hnf1ba* is not expressed in the CS gland.
DOI: https://doi.org/10.7554/eLife.38911.008

**Figure supplement 2.** Hnf1b nuclear export at 50 hpf.
DOI: https://doi.org/10.7554/eLife.38911.009

## Irx3b inhibits CS cell fate commitment in the pronephros

The formation of the DE segment has previously been shown to be regulated by the Irx3b transcription factor, which acts downstream of Hnf1b (*Wingert and Davidson, 2011*). We therefore explored the involvement of Irx3b in CS gland formation in loss-of-function studies. We found that *irx3b* knockdown using morpholinos did not prevent the initial formation of the DE segment, as *slc12a1*

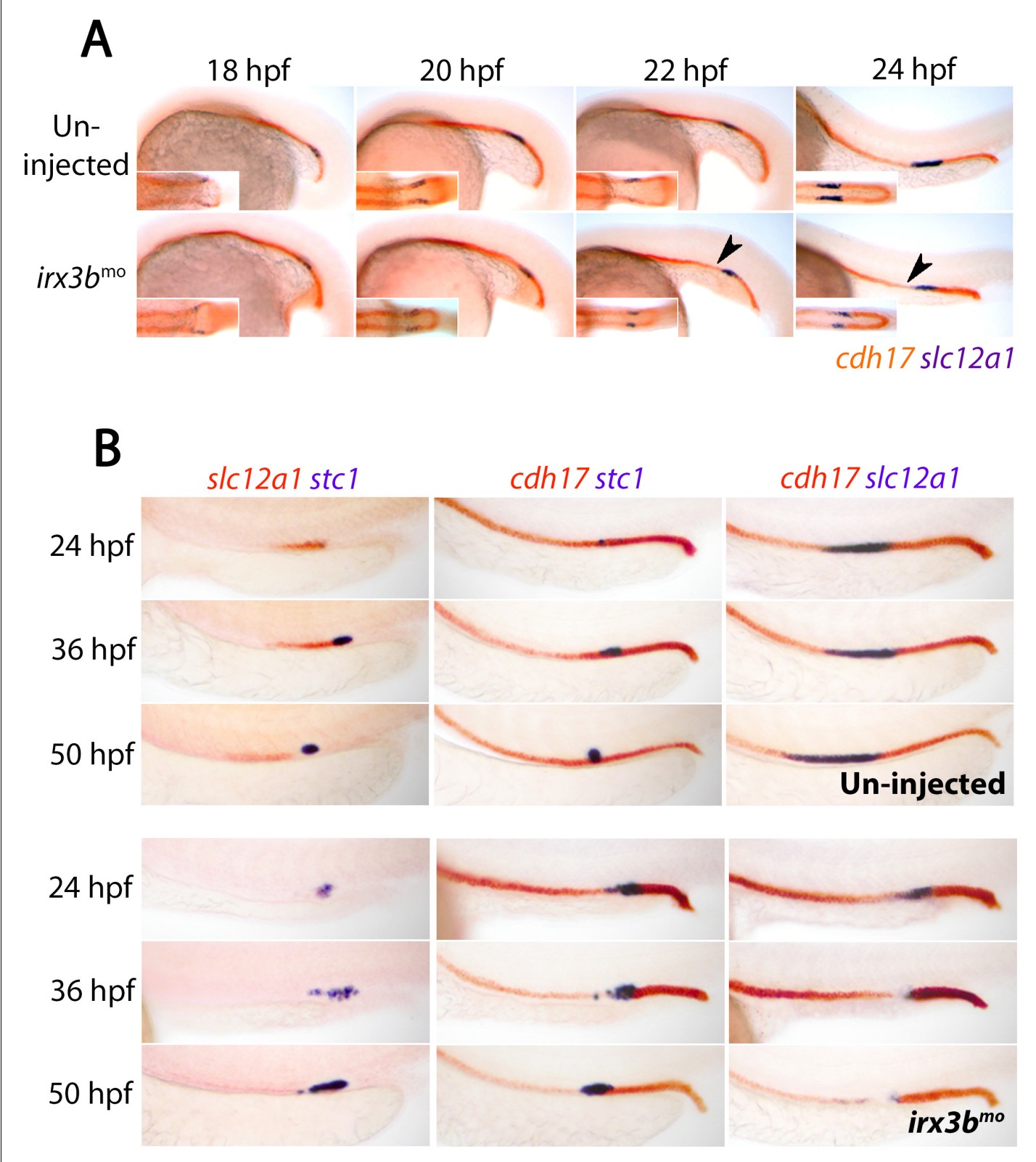

**Figure 4.** *irx3b* depletion converts the DE segment to CS fate. (**A**) Panels show lateral views of 18 hpf, 20 hpf, 22 hpf and 24 hpf wild-type and *irx3b* morphant embryos co-stained for *cdh17* (red) and *slc12a1* (purple). Black arrowheads highlight the initiation of pronephric phenotypes in *irx3b* morphants. (**B**) Panels show lateral views of control (un-injected) embryos and *irx3b* morphants double-stained for either *slc12a1*(red)/*stc1*(purple), *cdh17*(red)/*stc1*(purple) or *cdh17*(red)/*slc12a1*(purple) at 24 hpf, 36 hpf and 50 hpf.

*Figure 4 continued on next page*

*Figure 4 continued*

DOI: https://doi.org/10.7554/eLife.38911.011

displays a normal expression pattern in *irx3b* morphants at 18 hpf (n = 21/21) and 20 hpf (n = 24/24, *Figure 4A*). However, we noted that the length of the DE segment was reduced in some *irx3b*-deficient embryos at 22 hpf (n = 14/23) and at 24 hpf (n = 19/25, *Figure 4A*) compared to control embryos (n = 0/21 at 22 hpf and n = 0/43 at 24 hpf) and there was a corresponding expansion in *stc1*[+] CS cells from 24 hpf onwards (*Figure 4B*, *Table 1A and B*). By 36 hpf, the DE region lacked transcripts for *slc12a1*, consistent with previous reports (*Wingert and Davidson, 2011*) and *cdh17* was down-regulated in the region where ectopic *stc1*[+] cells are present (*Figure 4B*, *Table 1B and C*). An analysis of live *Tg(cdh17:egfp)* embryos that are deficient in *irx3b* showed that these ectopic CS cells still underwent apical constriction but were not fully extruded from the tubule (*Figure 5A*). In addition, we found that the tubule cells rostral to the enlarged CS gland display a stretched morphology in *irx3b*-deficient animals, both in live *Tg(cdh17:egfp)* embryos (*Figure 5A*) and by expression analysis of the pan-tubule marker *atp1a1a.4* (*Figure 5B*). Conversely, the DL segment was compacted and shorter in *irx3b*-depleted animals (*Figure 5A and B*).

Given that the tubule starts moving rostrally by 29 hpf (*Vasilyev et al., 2009*), the morphological effects caused by *irx3b* knockdown suggests that an enlarged and un-extruded CS gland may prevent the DL segment from undergoing this movement, resulting in greater stretching of the more proximal tubule cells and compaction of the distal tubule. To test this, *irx3b*-deficient animals were treated with Blebbistatin to block apical constriction. The majority of these animals were found to display relatively normal tubule morphology with ectopic *stc1*[+] cells scattered throughout the tubule (n = 20/25, *Figure 5C*). Taken together, these results suggest that Irx3b inhibits the transdifferentiation of DE cells to a CS fate and that the failed extrusion of the ectopic CS cells seen in *irx3b*-deficient embryos alters tubular cell morphology due to a block in the rostral migration of the DL segment.

**Table 1.** Empirical results for the effect of *irx3b* morpholino on *cdh17*, *stc1* and *slc12a1* expression at 24 hpf, 36 hpf and 50 hpf Un-injected wild-type embryos showed no phenotypes in the pronephros.
The phenotypes for *irx3b* morphants are: *slc12a1*, reduction in staining; *stc1*, increased/ectopic staining; *cdh17*, expanded DL domain, reduced DE staining and reduced/stretched PST segment. Experiments were repeated multiple times (at least three times) and the results showed are from one representative experiment.

| *slc12a1/stc1* | 24 hpf | 36 hpf | 50 hpf |
| --- | --- | --- | --- |
| Un-injected | 33 | 34 | 33 |
| *irx3b*[mo] *slc12a1* phenotype | 17/23 | 27/32 | 30/39 |
| *irx3b*[mo] *stc1* phenotype | 18/23 | 27/32 | 33/39 |
| *cdh17/stc1* | 24 hpf | 36 hpf | 50 hpf |
| Un-injected | 27 | 30 | 31 |
| *irx3b*[mo] *cdh17* phenotype | 25/27 | 29/31 | 30/38 |
| *irx3b*[mo] *stc1* phenotype | 22/27 | 26/31 | 31/38 |
| *cdh17/slc12a1* | 24 hpf | 36 hpf | 50 hpf |
| Un-injected | 23 | 23 | 27 |
| *irx3b*[mo] *cdh17* phenotype | 24/27 | 29/30 | 32/35 |
| *irx3b*[mo] *slc12a1* phenotype | 26/27 | 27/30 | 34/35 |

DOI: https://doi.org/10.7554/eLife.38911.010

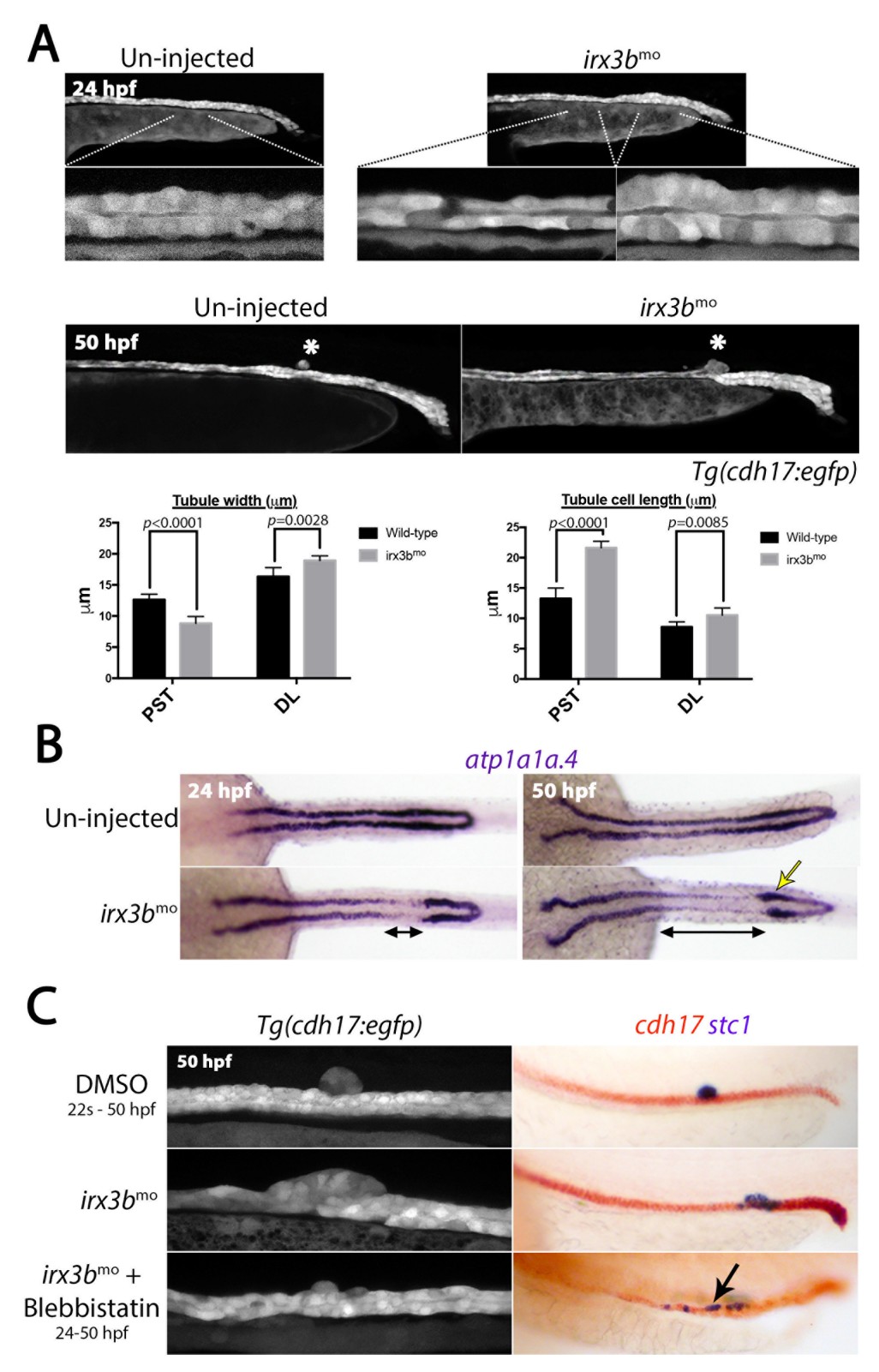

**Figure 5.** *irx3b* depletion causes aberrant pronephric morphogenesis. (**A**) Lateral views of live un-injected and *irx3b* morphant *Tg(cdh17::egfp)* embryos with lower panels showing higher magnification of indicated pronephric regions. Histograms show the width of the tubule (left) and length of tubule cells (right) in the PST and DL regions of wild-type and *irx3b* morphants (n = 10). (**B**) Panels show dorsal view images of embryos stained for *atp1a1a.4* at the stages indicated. Double-ended black arrows highlight areas of reduced *atp1a1a.4* expression immediately anterior to a thickened distal tubule

*Figure 5 continued on next page*

*Figure 5 continued*

(yellow arrow) at 24 hpf and 50 hpf in *irx3b* morphants. (C) Lateral views of the trunk of *Tg(cdh17:egfp)* embryos (left panels) and embryos double stained for *cdh17* (red) and *stc1* (purple; right panels) after the indicated treatments. Black arrow highlights the ectopic *stc1*[+] cells that remain in the tubule after *irx3b* knock down and Blebbistatin treatment.

DOI: https://doi.org/10.7554/eLife.38911.012

## Irx3b suppresses CS cell formation

We then went on to examine a link between Irx3b and Notch, as the Irx factors have been implicated as both positive and negative regulators of Notch signalling (*Scarlett et al., 2015*; *Domínguez and de Celis, 1998*). In addition, Notch has recently been connected to CS gland development (*Drummond et al., 2017*) and Notch components are expressed in CS cells (*Figure 6—figure supplement 1*). We first analysed the *TP1:VenusPEST* Notch reporter line, which contains multiple RBP-Jk-binding sites in front of a minimal promoter (*Ninov et al., 2012*), and found Notch activity in CS cells from 36 hpf onwards (*Figure 6A*). This result is consistent with the Notch pathway playing a role in CS gland formation. We did not observe fluorescence in presumptive CS cells or tubule multiciliated cells (which have previously been shown to form in a Notch-dependent manner (*Liu et al., 2007*; *Ma and Jiang, 2007*) prior to 36 hpf (*Figure 6—figure supplement 2*), but this may reflect low sensitivity of the reporter.

To assess the functional consequences of loss of Notch signalling, we treated wild-type zebrafish embryos with the γ-secretase inhibitor compound E (cpdE, *Figure 6B*) or DAPT (*Figure 6—figure supplement 3A*) from 18 hpf onwards and found reduced numbers of *stc1*[+] cells (n = 56/56 in cpdE treated, n = 29/31 in DAPT treated). However, as reported previously (*Drummond et al., 2017*), treatment with cpdE (n = 21/29) or DAPT (n = 20/34) from 10 hpf onwards paradoxically increased the size of the CS gland (*Figure 6—figure supplement 3B*), indicating that perturbation of Notch signalling has complex temporal effects on CS formation.

We next performed epistasis experiments to uncover if Notch signals are required upstream or downstream of *irx3b*. In our hands, we found that the effective dose of the *irx3b* morpholino was slightly toxic to the embryo, suggesting it may have off-target effects. These effects were pronounced in embryos treated with γ-secretase inhibitors (data not shown). To minimise these effects and to also confirm the specificity of the *irx3b* morpholino knockdown phenotype, we utilised the CRISPR-Cas9 system to introduce mutations into the *irx3b* gene (*Gagnon et al., 2014*; *Burger et al., 2016*). Injection of two gRNAs targeting *irx3b* reproduced the *irx3b* morpholino phenotype in $F_0$ animals (crispants) by increasing the number of *stc1*[+] cells in the DE region of the tubule (n = 45/49, *Figure 6B*). T7 endonuclease 1 analysis (*Huang et al., 2012*) for DNA mismatches in *irx3b* crispants indicated successful mutagenesis was achieved (*Figure 6—figure supplement 4A*). These crispant embryos were also grown to adulthood and used to generate stable mutant lines with a frame-shift deletion allele that similarly phenocopied the morphant/crispant phenotypes (*Figure 6—figure supplement 4B and C*). We found that *irx3b* crispants treated with cpdE from 18 to 50 hpf showed a robust reduction in the number of *stc1*[+] cells (n = 37/37; *Figure 6B*). Thus, we conclude that Irx3b acts upstream or in parallel to late (18 hpf onwards) Notch signalling to inhibit the transdifferentiation of DE cells into CS cells.

We next analysed the effects of *irx3b* depletion and γ-secretase inhibition on the intracellular localisation of Hnf1b (*Figure 6B*). We found in *irx3b* crispants at 50 hpf that the number of cells with nuclear-localised Hnf1b was quantitatively decreased in the region of the DE segment that shows unextruded CS cells (*Figure 6C*). In contrast, in wild-type (n = 27/27) or *irx3b* crispant embryos (n = 25/25) that had been treated with cpdE Hnf1b was retained in the nucleus of renal tubule cells in the DE region (*Figure 6B and C*). Treatment with cpdE also reduced the *TP1:VenusPEST* fluorescence in control (n = 28/28) and *irx3b* crispants (n = 32/45) consistent with the inhibitory effects of cpdE on CS cell formation being mediated by a block in Notch-Rbpj signal transduction (*Figure 6B and D*). Together, these results support a model in which Notch signalling and the cytoplasmic sequestration of Hnf1b in a subset of DE epithelial cells leads to the transdifferentiation of renal epithelial cells into CS cells.

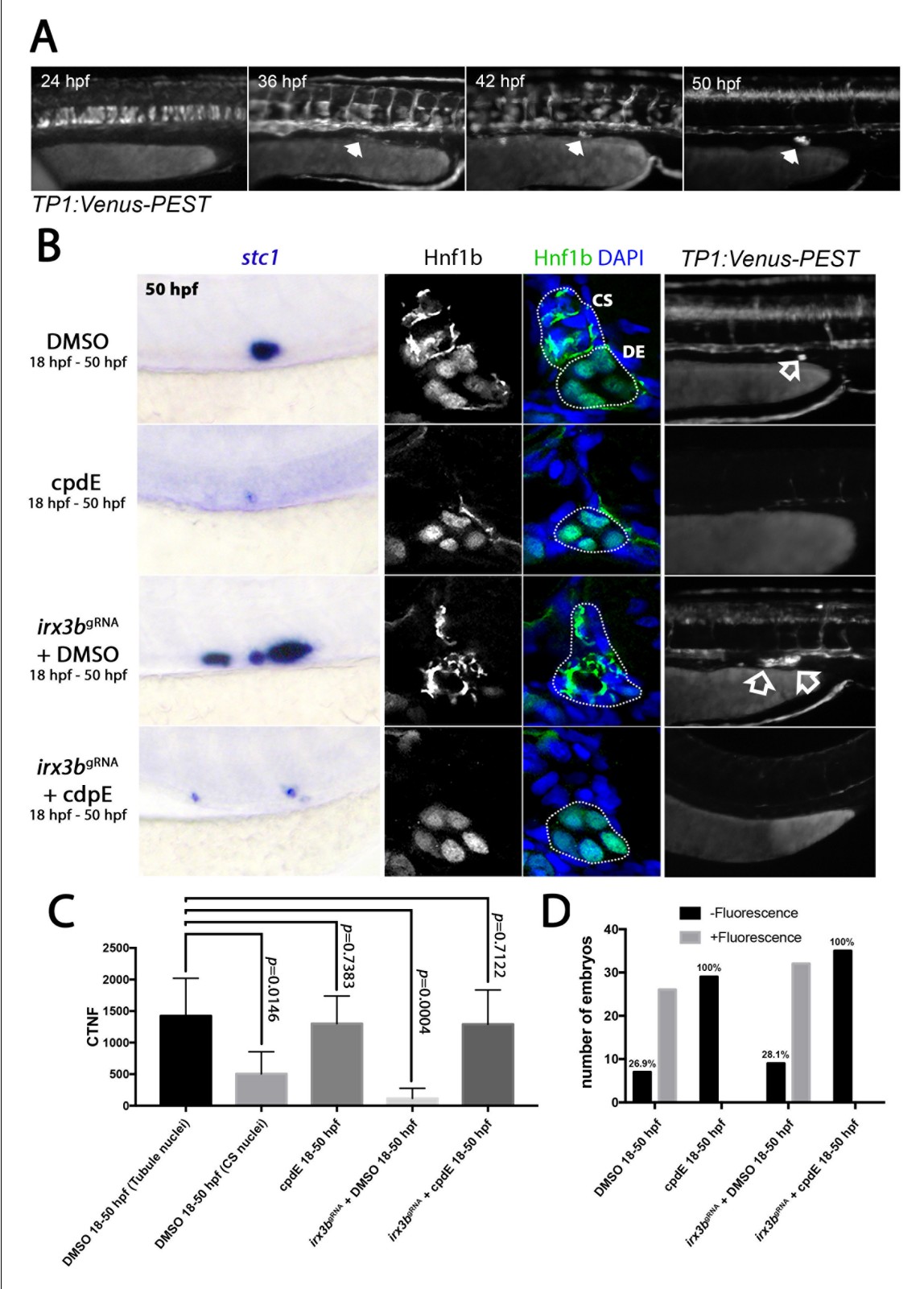

**Figure 6.** Notch inhibition precludes CS gland formation. (**A**) Panels show a stage series of fluorescent images of the *TP1:VenusPEST* transgenic line in the trunk region (arrows mark fluorescence in the region of the CS gland). (**B**) Left panels show lateral views of the posterior trunk showing *stc1* expression after the indicated treatments. Middle panels show transverse sections of the pronephric tubule and position of the CS gland showing Hnf1b staining (±DAPI) after the same treatments. Right panels show lateral views of the trunk in the region of the CS gland (highlighted by unfilled

*Figure 6 continued on next page*

*Figure 6 continued*

arrow). (**C**) Histogram showing 'corrected total nuclear fluorescence' (CTNF) measurements of nuclei in the positions indicated. (**D**) Histograms showing the number of embryos with fluorescence in the CS gland in progeny from *TP1:VenusPEST* in-crosses with the indicated treatments.

DOI: https://doi.org/10.7554/eLife.38911.013

The following figure supplements are available for figure 6:

**Figure supplement 1.** Notch components are expressed in the CS gland.

DOI: https://doi.org/10.7554/eLife.38911.014

**Figure supplement 2.** Notch activity in the pronephros is not observed early in development in the TP1:VenusPEST reporter line.

DOI: https://doi.org/10.7554/eLife.38911.015

**Figure supplement 3.** Early Notch inhibition promotes CS gland formation but later treatment inhibits CS development.

DOI: https://doi.org/10.7554/eLife.38911.016

**Figure supplement 4.** *irx3b* crispants and stable mutants recapitulate the renal phenotypes associated with *irx3b* morpholino knock down.

DOI: https://doi.org/10.7554/eLife.38911.017

# Discussion

In this study, we have discovered a rare example of naturally occurring transdifferentiation that arises in the zebrafish embryonic kidney and results in the formation of endocrine cells from a differentiated renal tubular epithelium. By live-cell imaging, we show that the CS cells are extruded from the tubule via apical constriction and mass cell extrusion. We investigated the molecular pathways governing this transdifferentiation event and identified the Notch pathway as being a critically important signal together with cytoplasmic sequestration of Hnf1b, a master determinant of renal epithelial fate. The Irx3b transcription factor was found to play an inhibitory role in this process by preventing widespread transdifferentiation of the DE segment. Together, these results establish a new model system to understand the molecular and cellular processes governing cellular plasticity in vivo.

Our data indicate that the transdifferentiation of DE to CS fate occurs in sequential steps and the finding that CS cell numbers are reduced by pharmacological inhibition of Notch signalling from 18 hpf onwards, just prior to the appearance of *stc1*[+] cells, suggests that the Notch pathway plays an early instructive role. In support of this, we found evidence of Notch-Rbpj signalling in CS cells using the *TP1:Venus-PEST* reporter, although the earliest we could detect fluorescence was 36 hpf, perhaps reflecting the delay needed to accumulate enough VenusPEST protein to be detectable. Interestingly, we also failed to see Venus-PEST fluorescence in developing multi-ciliated cells that are induced by Notch around 16 hpf (*Liu et al., 2007*; *Ma and Jiang, 2007*), therefore the reporter may not be sensitive enough to detect all Notch signalling.

In conflict to the notion that Notch induces CS fate, global overexpression of *nicd1*, encoding the constitutively active signalling moiety of the Notch one receptor, has been reported to decrease CS cell number, whereas treatment with the Notch inhibitors DAPT or cpdE (our study) starting at the end of gastrulation increases CS cell number (*Drummond et al., 2017*). The molecular basis for these contradictory outcomes is not clear and needs more investigation. It is possible that Notch signalling is serving multiple functions at different times of development, as found in other contexts (*Guruharsha et al., 2012*). For instance, early perturbation of Notch signalling may lead to the renal cells being more competent to transdifferentiate at later stages, either by overcoming a requirement for Notch later, or by lowering their threshold to respond so that they are exquisitely sensitive at later stages, despite the Notch pathway being dampened in the presence of an inhibitor.

Notch has well-described roles in determining binary cell fate decisions during progenitor cell decisions including in the zebrafish pronephros where it acts by a classical lateral-inhibition mechanism to inhibit a multi-ciliated phenotype in renal progenitors (*Liu et al., 2007*; *Ma and Jiang, 2007*; *Marra and Wingert, 2016*). A similar lateral-inhibition pathway may be required for the DE-to-CS transdifferentiation but with the timing of this occurring after the multi-ciliated cell fate decision (*Wingert et al., 2007*; *Liu et al., 2007*; *Ma and Jiang, 2007*). We, and others have found that a number of Notch signalling components are expressed in the pronephric tubules during the period that the CS gland arises (*Wingert et al., 2007*; *Liu et al., 2007*; *Ma and Jiang, 2007*). A comprehensive loss-of-function analysis of these genes is needed in order to determine which of these is involved in CS formation. In the context of transdifferentiation, work in *C. elegans* has implicated Notch in the conversion of rectal epithelial Y cells into motor neurons, although in this case the Y cell

transitions to an intermediary cell state rather than directly transdifferentiating as seen in our study (*Jarriault et al., 2008*). In a mammalian example, bronchial cells known as 'club' cells, which depend on active Notch for their maintenance, switch to a ciliated phenotype following disruption of Notch signalling in adult mice, without an apparent dedifferentiation stage or need for cell division (*Lafkas et al., 2015*). In this example, 'club' cells and ciliated cells share a common progenitor during lung development whereas in our study the renal and glandular lineages have, until now, been considered to be developmentally distinct fates. Indeed, the conversion we see of a renal epithelial cell becoming an endocine gland can be considered more of a trans-organogenesis event.

Mechanistically, how Notch signalling induces the DE to CS fate switch requires more study. Our observation that the Hnf1b transcription factor, a master regulator of renal epithelial fate, becomes sequestered in the cytoplasm of cells undergoing transdifferentiation suggests that Notch regulates the levels of Hnf1b in the nucleus either by inducing Hnf1b nuclear export or alternatively, if the nuclear turnover of Hnf1b is high, by preventing newly translated Hnf1b from entering the nucleus. We favour the former notion as Hnf1b is considered a relatively stable 'bookmarking' factor that remains associated with chromatin during cell division where it reinstates transcriptional programs following mitotic exit (*Verdeguer et al., 2010*). As Hnf1b is critical for renal tubule differentiation in zebrafish and mice (*Naylor et al., 2013*; *Heliot et al., 2013*; *Massa et al., 2013*; *Naylor and Davidson, 2014*), we hypothesise that there is an on-going requirement for Hnf1b in the tubule to maintain renal epithelial identity and that sequestration of this factor is sufficient to induce the transdifferentiation of DE cells into CS cells. In support of this, we found that *hnf1b*-deficient embryos show ectopic *stc1* expression along a broad stretch of the pronephric tubule, consistent with a CS program being activated in these cells. Whether CS identity is a default state in the tubule is unclear, but we have observed that the Notch ligand *jagged-2a* is abnormally maintained in the pronephric tubule of *hnf1b*-deficient embryos (data not shown) consistent with aberrant Notch signalling also being involved in the induction of ectopic *stc1*. The notion that sequestration of a master transcription factor can induce transdifferentiation provides a new paradigm for the field, as most studies of cell fate plasticity have reported a need for regulatory factors to be up-regulated in order to drive the process, particularly those involving non-physiological 'reprogramming' (reviewed in [*Daniel et al., 2016*]).

Our data point to the Irx3b transcription factor as playing an important role in the DE-to-CS fate switch. We have previously linked Hnf1b and Irx3b together by demonstrating that *hnf1b*-deficient embryos fail to express *irx3b* in the renal tubules (*Naylor et al., 2013*), placing *hnf1b* upstream of *irx3b* in a genetic hierarchy. In this study we show that *irx3b* deficiency does not influence the initial specification of the DE segment as previously thought (*Wingert and Davidson, 2011*) but instead leads to the dramatic transdifferentiation of this segment into CS cells. Consistent with our notion that Hnf1b sequestration drives this transdifferentiation, we found that the ectopic CS cells display cytoplasmic Hnf1b. Furthermore, inhibiting Notch signalling rescued the CS expansion phenotype in *irx3b*-deficient animals, including Hnf1b nuclear localisation, placing Irx3b upstream or in parallel to the Notch pathway. Irx/Iroquois family transcription factors have been reported to modulate Notch signalling in other cellular contexts. Most notably during *Drosophila* eye development, where the Iroquois genes establish where Notch activation occurs by controlling the levels of Fringe, a glycosyl-transferase that alters the sensitivity of the Notch receptors to transactivation from the different classes of Notch ligand (*Panin et al., 1997*; *Moloney et al., 2000*). In vertebrates, including zebrafish, there are three Fringe homologues (*Qiu et al., 2004*) but their expression patterns during CS formation have not been closely examined. One speculative idea is that Irx3b modulates Notch signalling in the DE segment via the regulation of *fringe* or other Notch pathway genes, to ensure that the CS-inducing effects of Notch signalling are suppressed. This may occur by promoting one Notch-Ligand pairing over another or by altering the strength of the Notch signal. It is also interesting to note that cross-regulatory interactions between Notch and Irx3 have been shown in human microvascular endothelial cells where *Irx3* induces expression of *Delta-like ligand four* but is in turn repressed by the Notch mediator Hey1 via direct binding to the *Irx3* promoter (*Scarlett et al., 2015*). How the proposed repressive effects of Irx3b on Notch signalling are overcome so that the DE-to-CS transdifferentiation event can be initiated at the appropriate time during development remain to be determined in future studies. Similarly, how the Notch, Irx3b and Hnf1b factors interact with other transcriptional regulators that have been reported to control CS cell number, such as Sim1a and Tbx2a/b (*Cheng and Wingert, 2015*; *Drummond et al., 2017*), also need investigating.

The second key step in the transdifferentiation of DE cells to a CS gland is the expulsion of the fate-changed cells from the basal side of the tubule. This process presents a challenge to the embryo as any loss of cells from the epithelium has the potential to compromise the barrier function of the tubule and cause lumenal fluid to leak out into the interstitium. Our live-cell imaging revealed that the first step in achieving this feat is the constriction of the apical membranes of the CS cells. Apical constriction is a well-known feature of polarised epithelia that drives bending, folding and tube formation during the formation of different tissues and organs (*Sawyer et al., 2010*; *Martin et al., 2009a*; *Mason et al., 2013*; *Sadler, 2005*; *Martin and Goldstein, 2014*). Apical constriction is driven by contraction of an apical meshwork of F-actin by the molecular motor myosin IIA (*Martin et al., 2009b*) and consistent with this, we found that blebbistatin treatment inhibited CS cell extrusion. Interestingly, in the *Drosophila* eye-antennal disc, Notch activation has been found to trigger actomyosin-mediated cell apical constriction, raising the possibility that the Notch pathway may both induce DE-to-CS transdifferentiation and initiate apical constriction (*Ku and Sun, 2017*).

In the developing avian lung, where the tubular epithelium of the primary bronchus undergoes apical constriction in localised areas, the result is the formation of lung buds that then grow out as new airways (*Locy WA, 1916*). Unlike this system, the 'bud' of CS cells we observe is made up of far fewer cells, does not transition to a tubular outgrowth and proliferation is not required for its extrusion, although we do observe CS cells proliferating once they are free from the tubule. Instead, the budding CS cells are 'shed' basally, *en masse*, as an intact ball of cells. Thus, we conclude that while apical constriction initiates the process, additional cellular behaviours must then be required to extrude the cells. One possible mechanism for CS cell extrusion is that the CS cells undergo EMT in order to disassemble their tight junctions with their neighbours. This is an attractive hypothesis, particularly given that this process operates in other related tissues such as the pancreas, where the primitive duct epithelium responds to low Notch signalling to down-regulate Cdh1 in single cells that then undergo EMT and delaminate as endocrine precursors (*Shih et al., 2012*). However, we do not observe a loss of Cdh1, which is a widely used definition of EMT (*Kalluri and Weinberg, 2009*). Furthermore, from our histological analysis we have not been able to capture an intermediary mesenchyme-like state and observe no migration of individual CS cells. Instead, we find that CS cells rapidly and seamlessly change from a cuboidal epithelium in the renal tubule to a simple lobular epithelium of the CS gland. Based on this, we favour a model in which CS cells are extruded by a non-EMT-based pathway that may be more comparable to the epithelial remodelling that occurs during primary spinal cord formation in the embryo. Here, the neural plate undergoes apical constriction and invaginates before pinching off as the neural tube while the surrounding non-neural ectoderm fuses together (*Nikolopoulou et al., 2017*). The exact molecular mechanism for how barrier function is maintained during neural tube closure is not fully understood but epithelial-to-mesenchymal (EMT) transitions appear suppressed (*Ray and Niswander, 2016*).

Prior work has shown that overcrowding of epithelial sheets results in live cell extrusion of single cells by a mechanism that involves the formation of an actomyosin contractile ring in cells surrounding the extruded cell. This ring then contracts and the innermost cell is pushed out of the epithelium (*Eisenhoffer et al., 2012*; *Gu et al., 2011*). We performed laser ablations immediately anterior and posterior to the CS gland to disrupt such a ring in neighbouring cells but this failed to prevent CS extrusion. In addition, we have found that treating embryos with the $S1P_2$ receptor inhibitor JTE-013 has no effect on CS extrusion (data not shown). Thus, it is unlikely that this mechanism of live cell extrusion is playing a role in CS gland shedding.

Taken together, our data show that the conversion of DE cells to a CS gland fate in the zebrafish renal tubule represents a novel example of a developmentally programmed direct transdifferentiation event coupled to an unusual live cell extrusion behaviour. This model provides new ways to interrogate the plasticity of differentiated cellular states, will advance efforts to harness transdifferentiation for therapeutic applications, and furthers our understanding of epithelial cell behaviours.

## Materials and methods

### Zebrafish husbandry

Zebrafish embryos were maintained and staged according to established protocols (*Kimmel et al., 1995*) and in accordance with the University of Auckland's Animal Ethics Committee (protocol

001343). Paired matings of wild-type *Tübingen* (*Tg*), transgenic *Tg(cdh17:egfp)*, and mutant *Tübingen*[+/hi1843] adults (RRID_ZFIN_ZDB-GENO-090923–3) were carried out in order to collect embryos used to perform the experiments described.

## MO design and injection

Previously validated morpholinos to *irx3b* (5′- ACCGGGAGGACTGCGGGGAAACTCG −3′ (*Wingert and Davidson, 2011*) and *hnf1bb* (5′-CTTGGACACCATGTCAGTAAA-3′ (*Choe et al., 2008*) were purchased from Gene Tools LLC and re-suspended in 1X Danieau solution. One-cell embryos were injected with 1–5 nl of morpholino at 1.5 ng/nl for *irx3b* and 5 ng/nl for *hnf1bb*. Embryos were incubated at 28.5 °C to the desired stage and fixed in 4% paraformaldehyde. All experiments were repeated at least three times to reproducible yield the phenotypes as described in the text.

## gRNA design and injection

gRNAs were designed using CHOPCHOP (Harvard) software and synthesised according to published protocols (*Gagnon et al., 2014*). We modified this protocol slightly to create two gRNAs that simultaneously targeted the first exon (5′-GGCGCGGAGATCTCGGTCAC-3′) and second exon (5′-GGA TGCAGAAAAACGAGATG-3′) of the *irx3b* locus in order to increase the chance of biallelic gene disruption. SP6 promoter containing gRNA templates were generated using Phusion polymerase (ThermoFisher). Transcripts were synthesised using the SP6 Megascript kit (Ambion). Cas9 protein that included a nuclear localisation signal was purchased from PNA-BIO Inc. (CP02). Approximately 10 nl of 0.2 ng/μl *irx3b* gRNAs[1+2], 0.4 ng/μl Cas9 and 300 mM KCl were injected directly into the cell of early one-cell stage embryos as per the *Burger et al. (2016)* protocol. For the T7 endonuclease assay, five un-injected and five *irx3b* gRNAs injected embryos were lysed in 50 μl of 50 mM NaOH at 95°C for 20 min. 5 μl of 100 mM Tris HCl pH 8.5 was added to the lysed embryos and the solution was homogenised. 1 μl of the lysed embryos containing genomic DNA was used in a PCR reaction to amplify *irx3b* exon 1, with PCR primers flanking the site of mutagenesis (Forward primer 5′-TCCCGCAGCTAGGCTATAAGTA-3′; Reverse primer 5′- ACGGGATCAAATCTGAGCTATT-3′). After purification of the PCR product, 200 ng of amplified DNA, 2 μl of 10X NEB buffer and water (to a final volume of 19 μl) was mixed together and placed in a thermocycler for rehybridisation; 5 min at 95°C, ramp down to 85°C at −2°C/s, ramp down to 25°C at −0.1°C/s, 25°C for 10 min. 1 μl of T7 Endonuclease 1 (NEB) was then added to the rehybridisaed PCR product and incubated at 37°C for 20 min before the products were ran on a 1.5% agarose gel to analyse for digestion.

## Drug treatments

Embryos were dechorionated and exposed to pharmacological agents at the time-points outlined in the text. Blebbistatin (SigmaAldrich #B0560), DAPT (Generon, A8200) and Compound E (Abcam, ab142164) were dissolved in DMSO to a stock concentration of 10 mM and diluted in E3 media to a working concentration of 10 μM, 100 μM and 50 μM, respectively. Aphidicolin (SigmaAldrich #A0781) was dissolved to a 10 mM stock solution with DMSO. The combined Hydroxyurea (20 mM) and Aphidicolin (150 μM) solution (termed HUA in the text) was made up in E3 embryo medium.

## Whole mount in situ hybridisation and immunohistochemistry

Whole mount in situ hybridisation was performed using protocols previously described (*Thisse and Thisse, 2008*). Digoxigenin and Fluorescein anti-sense riboprobes were synthesised using T7/T3/SP6 RNA polymerase transcription kits (Roche Diagnostics) from plasmids used previously (*Wingert and Davidson, 2011*; *Wingert et al., 2007*). For double in situ hybridisations, alternative Dig- or Flu-riboprobes were used and alkaline phosphatase was inactivated by two 15 min 100 mM Glycine (pH 2.2) treatments. To perform whole mount antibody staining, fixed embryos were washed twice in PBS containing 0.05% Tween20 (PBST), then placed in PBS containing 0.5% Triton X100 for 20 min to permeabilise the embryo. Embryos were then washed twice in TBS containing 0.05% Triton X100 (TBST) and blocked in TBST containing 3% BSA and 5% Goat serum for at least 1 hr. The Hnf1b (SantaCruz Ab #22840, RRID:AB_2279595 (discontinued) and Sigma-Aldrich #HPA002803, RRID:AB_1080232) and E-cadherin (SantaCruz Ab #7870, RRID:AB_2076666) primary antibodies was added at 1:500 dilution overnight at 4°C. Embryos were washed five times in PBST before Goat anti-Rabbit

IgG DyLight594 conjugated secondary antibody (Abcam #96901, RRID:AB_10679699) was added at 1:500 dilution. Embryos were incubated in the secondary antibody for 2 hr at room temperature, washed twice in PBST then imaged on a FV1000 LiveCell confocal microscope (Olympus).

### Corrected total nuclear fluorescence measurements

Corrected total nuclear fluorescence (CTNF) measurements (as shown in *Figures 3* and *6*) were determined using Fiji to detect Nucleus area, Nucleus Mean Fluorescence and average background fluorescence. CTCF was calculated using the formula; (*Nucleus area x Nucleus mean fluorescence*) – (*Nucleus Area x Average background fluorescence*)

### Cryosectioning

Zebrafish embryos were fixed in 4% PFA overnight at 4°C and washed twice in PBS and then placed in 1% low melting point agarose containing 5% sucrose and 0.9% agar (made up in water) in cryomoulds. After 1 hr, the cryoblocks were removed from the cryomoulds and placed in 30% Sucrose/PBS. Cryoblocks were incubated overnight at 4°C then 30% Sucrose/PBS was replaced and left for another night at 4°C. Cryoblocks were flash frozen on dry ice and immediately sectioned. 14 µm sections were cut in a cryostat machine set to −25°C and transferred to gelatin-coated microscope slides.

### EdU assay

For exposure to the EdU label, *Tg(cdh17:egfp)* embryos were injected with 0.5 µg/ml EdU (Santa-Cruz #284628). Injections were performed into the yolk (embryo was immobilised in a well formed from a 1% agarose bed in a 45 mm petri dish) and left in E3 embryo medium for the periods of development outlined in the text (we find incubating embryos in E3 media containing EdU only permitted superficial incorporation at post gastrula stages). Embryos were fixed at the desired stage and then incorporated EdU was detected using a Click-iT EdU Alexa Fluor 594 kit (Life Technologies). We found that this labelling quenched the endogenous eGFP fluorescence in the transgenic embryo, thus antibody staining using the same protocol described above was performed for eGFP (SantaCruz Ab #8334, 1:500).

### Time-lapse imaging

32 hpf *Tg(cdh17:egfp)* embryos were anaesthetised in embryo media containing 0.02% Ethyl 3-aminobenzoate methanesulfonate (Tricaine) and 0.003% N-phenylthiourea (PTU). Anaesthetised embryos were placed in a well made in a 1.5% agarose gel and positioned appropriately for image capture. Z-stack confocal images were captured every 20 min for 20 hr on a FV1000 LiveCell confocal microscope (Olympus) to generate the time-lapse video.

### Laser ablation

Our laser ablation protocol was similar to that described previously (*Naylor et al., 2016a*; *Palmyre et al., 2014*). *Tg(cdh17:egfp)* embryos at 32 hpf were manually de-chorionated and placed into a well on a petri dish containing a bed of 1% agarose. Embryos were oriented to be positioned laterally and GFP-labelled pronephric tubule at the position of the presumptive CS gland was photoablated using a 60X water-dipping objective. We found a 5 min exposure to a 405 nm diode laser at maximum intensity with a 12.5 µs/pixel dwell time was required to enable severance of the tubule, when observed later at 50 hpf. The specific target region corresponding to a circular area with a diameter of 25 µm was selected using a Sim Scanner on an Olympus FluoView FV1000 confocal laser-scanning microscope. Subsequent to photo-ablation, embryos were transferred to fresh E3 embryo medium and grown to 50 hpf for further analysis.

## Acknowledgements

We thank S Patke for managing and taking care of the fish. This work was supported by a Health Research Grant of New Zealand 13/332 to Alan J Davidson.

## Additional information

### Funding

| Funder | Grant reference number | Author |
|---|---|---|
| Royal Society of New Zealand | RSNZ/JSP-UOA1401-JR | Alan J Davidson |
| Health Research Council of New Zealand | HRC 15/057 | Alan J Davidson |
| Health Research Council of New Zealand | 13/332 | Alan J Davidson |

The funders had no role in study design, data collection and interpretation, or the decision to submit the work for publication.

### Author contributions

Richard W Naylor, Conceptualization, Resources, Data curation, Formal analysis, Validation, Investigation, Visualization, Methodology, Writing—original draft, Project administration, Writing—review and editing; Hao-Han G Chang, Sarah Qubisi, Data curation, Formal analysis; Alan J Davidson, Conceptualization, Resources, Data curation, Software, Formal analysis, Supervision, Funding acquisition, Validation, Investigation, Visualization, Methodology, Writing—original draft, Project administration, Writing—review and editing

### Author ORCIDs

Richard W Naylor  http://orcid.org/0000-0003-2901-7677
Hao-Han G Chang  http://orcid.org/0000-0001-8023-6862
Alan J Davidson  http://orcid.org/0000-0002-5732-1193

### Ethics

Animal experimentation: This study used zebrafish embryos, which were maintained and staged according to established protocols (Kimmel, Ballard, Kimmel, Ullmann, & Schilling, 1995) and in accordance with the University of Auckland's Animal Ethics Committee (protocol 001343).

### Decision letter and Author response

Decision letter https://doi.org/10.7554/eLife.38911.022
Author response https://doi.org/10.7554/eLife.38911.023

## Additional files

### Supplementary files

• Transparent reporting form
DOI: https://doi.org/10.7554/eLife.38911.018

### Data availability

The empirical counts for the experiments described in the manuscript are available as a spreadsheet on the Dryad database.

The following dataset was generated:

| Author(s) | Year | Dataset title | Dataset URL | Database and Identifier |
|---|---|---|---|---|
| Naylor RW, Davidson AJ | 2018 | Data from: A novel mechanism of gland formation in zebrafish involving transdifferentiation of kidney cells and live cell extrusion | http://dx.doi.org/10.5061/dryad.3hp84t0 | Dryad Digital Repository, 10.5061/dryad.3hp84t0 |

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
