## [Decision Letter]

Thank you for submitting your article "A novel mechanism of gland formation in zebrafish involving transdifferentiation of kidney cells and live cell extrusion" for consideration by *eLife*. Your article has been reviewed by Didier Stainier as the Senior Editor, a Reviewing Editor, and three reviewers. The following individual involved in review of your submission has agreed to reveal his identity: Benjamin M Hogan (Reviewer #1).

The reviewers have discussed the reviews with one another and the Reviewing Editor has drafted this decision to help you prepare a revised submission.

Summary:

Naylor and Davidson show an intriguing process whereby a subset of developing kidney epithelial cells transdifferentiate, extrude from the epithelial tubules and form the Corpuscles of Stannius (CS) – an endocrine gland. They suggest that the process is regulated by apical constriction within the epithelial tube and by a molecular pathway involving Hnf1, Irx3b and Notch signalling. The study combines some very nice live imaging approaches with careful genetic analyses using zebrafish knockdown, mutant and chemical inhibition. It is very well written, presents logically and is generally supported by high quality data.

The finding overall is of interest as it identifies a new trans-differentiation process in developing epithelia and opens up a new model to study this process.

However, there are several points where the depth of analysis does leave some room for doubt (such as the Notch signalling data, the issue of transdifferentiation - see below) and at many places there seemed to be a lack of quantitative data to support the observations being made. I outline below major and minor issues that should be addressed.

Essential revisions:

1) The claim that Notch signalling regulates CS trans-differentiation is supported by analysis of the mindbomb mutant (which has a constitutive loss of Notch signalling) and treatment with the gamma secretase inhibitor cpdE, treated at one timepoint. This is very limited to support the conclusions, especially considering that there are conflicting data previously published (Drummond et al., 2016).

a) The Discussion section suggests that DAPT treatment timing (from late gastrulation instead of 18 hpf used here) may have led to the wrong conclusions in the earlier Drummond et al., study. Can the authors show if DAPT from 18 hpf but not late gastrulation reproduces their data? Likewise, does early gastrulation treatment with cpdE expand the CS domain? Having both inhibitors and multiple stages analysed would improve confidence in the Notch claims.

b) Can the authors show that restricted Notch signalling precedes CS morphogenesis (e.g. using a Notch reporter transgenic line)? Can timing of Notch be correlated with Hnf1b exclusion from tubule nuclei?

c) The findings that mindbomb mutants do not generate CS is also opposite to the general findings of Drummond et al. One issue is that by 50hpf, the mindbomb mutant does not form a proper kidney and thus no CS cells can form in the absence of the DE tubules. It appears that the Cadh17 staining is thinner in mindbomb than in WT embryos (Figure 6A). Thus, the statement that Notch is involved in CS formation is misleading and maybe secondary to proper kidney development.

- Please provide further validation and experimental evidence to support conclusions made relating to the role of Notch signalling.

2) To claim "transdifferentiation" is somewhat premature. The main concern is that in the strictest sense, transdifferentation occurs when a fully differentiated, mature cell type changes to new cell type and lose all characteristics of the original cell. The main issue here is that the pronephric tubule cells may still be poised in a progenitor state. This study focuses on a process that occurs from 24hpf to 50hpf. This is within larval developmental stages and that many cells are likely to be still poised in a progenitor fate and have not completed differentiation. To sufficiently argue that this is transdifferentiation, one would have to perform a detailed analysis of progenitor markers versus differentiation markers of the kidney tubule.

- Please provide further evidence that this is indeed a transdifferentiation event by using further marker analysis.

3) At many points there is limited quantitative information for each experiment. For example;

a) Figure 1 and Figure 2 show individual representative images of IF and ISH stains to show how CS morphogenesis occurs. No n values, measurements or quantitative approaches are used to assign these behaviours and so it is difficult to know how robust these observations are. The authors should provide some form of quantitative analysis of the process.

b) Cdh1 staining in Figure 2A is stated to be largely lost at 50hpf. This is not clear in the image shown. Some quantification or at least demarcate the region where this loss of expression is seen should be included. Also, the authors state that F-actin is lost in CS cells at 50 hpf, but there is only one image at 50hpf and not earlier stages to compare with.

c) Throughout the manuscript n-values are only given for treatments and not for controls. Control numbers should also be provided and statistical analysis provided for all the main findings.

d) How many mindbomb embryos were counted and were stc1+ cells counted or is the analysis purely qualitative based on the size of the stc1 stained region?

e) The Irx3b mutants – no n-values are provided and only cdh17 expression is analysed. Please provide full details of analysis.

f) The authors mostly used non-quantitative approaches to assess the morphological phenotypes in Figures 4 and Figure 5.

- Please improve the use of quantification of data and statistics for these points and throughout.

4) The concept that Hnf1b is being sequestrated into the cytoplasm in CS forming cells is intriguing as this is one example of how cell fates can change. However, the data shown with antibody staining is not convincing as shown in Figure 3A, Figure 6C and in Figure 3—figure supplement 2.

There seems to be some inconsistency in the staining pattern of Hnf1b. The speckled pattern is apparent in the right panel of Figure 3A (50 hpf) and Figure 3—figure supplement 2, but not in the middle panel of Figure 3A (22 hpf) and Figure 6C, in which Hnf1b signals appear to be undetectable in CS cells.

In Figure 3—figure supplement 1, hnf1ba mRNA expression was not detected in CS cells. Does this suggest that Hnf1b pathway is inhibited in the transdifferentiated CS cells by transcriptional suppression rather than by cytoplasmic sequestration?

- Please improve clear quantification of the staining for Hnf1b and clarify if loss of expression or cytoplasmic sequestration is the major mechanism here.

---

## [Author Response]

Summary:Naylor and Davidson show an intriguing process whereby a subset of developing kidney epithelial cells transdifferentiate, extrude from the epithelial tubules and form the Corpuscles of Stannius (CS) – an endocrine gland. They suggest that the process is regulated by apical constriction within the epithelial tube and by a molecular pathway involving Hnf1, Irx3b and Notch signalling. The study combines some very nice live imaging approaches with careful genetic analyses using zebrafish knockdown, mutant and chemical inhibition. It is very well written, presents logically and is generally supported by high quality data.The finding overall is of interest as it identifies a new trans-differentiation process in developing epithelia and opens up a new model to study this process.However, there are several points where the depth of analysis does leave some room for doubt (such as the Notch signalling data, the issue of transdifferentiation -see below) and at many places there seemed to be a lack of quantitative data to support the observations being made. I outline below major and minor issues that should be addressed.Essential revisions:1) The claim that Notch signalling regulates CS trans-differentiation is supported by analysis of the mindbomb mutant (which has a constitutive loss of Notch signalling) and treatment with the gamma secretase inhibitor cpdE, treated at one timepoint. This is very limited to support the conclusions, especially considering that there are conflicting data previously published (Drummond et al., 2016).a) The Discussion section suggests that DAPT treatment timing (from late gastrulation instead of 18 hpf used here) may have led to the wrong conclusions in the earlier Drummond et al., study. Can the authors show if DAPT from 18 hpf but not late gastrulation reproduces their data? Likewise, does early gastrulation treatment with cpdE expand the CS domain? Having both inhibitors and multiple stages analysed would improve confidence in the Notch claims.

In the revised manuscript we have performed DAPT and cpdE treatments from 10 hpf to 50 hpf and 18 hpf to 50 hpf. We find that the earlier treatment (10-50 hpf) does indeed expand the *stc1* expression domain with both DAPT (as previously reported by Drummond et al.,) and cpdE (shown here), but later treatment (18-50 hpf) with either γ-secretase inhibitor reproduced our previous finding of reduced *stc1* expression. These results are shown in Figure 6—figure supplement 3 of the resubmission. We conclude from this additional data that the timing of Notch signals is imperative for appropriate transdifferentiation of DE tubule cells to CS cells and have the following statement in the revised text:

“To assess the functional consequences of loss of Notch signalling, we treated wild-type zebrafish embryos with the γ-secretase inhibitor compound E (cpdE, Figure 6B) or DAPT (Figure 6—figure supplement 3A) from 18 hpf onwards and found reduced numbers of *stc1*^+^ cells. However, as reported previously ^48^ treatment with cpdE or DAPT from 10 hpf onwards paradoxically increased the size of the CS gland (Figure 6—figure supplement 3B), indicating that Notch signalling is associated with complex temporal effects on CS formation.”

and

“In conflict to the notion that Notch induces CS fate, global overexpression of *nicd1*, encoding the constitutively active signaling moiety of the Notch 1 receptor, has been reported to decrease CS cell number, whereas treatment with the Notch inhibitors DAPT or cpdE (our study) starting at the end of gastrulation increases CS cell number ^48^. The molecular basis for these contradictory outcomes is not clear and needs more investigation. It is possible that Notch signaling is serving multiple functions at different times of development, as found in other contexts ^55^. For instance, early perturbation of Notch signaling may lead to the renal cells being more competent to transdifferentiate at later stages, either by overcoming a requirement for Notch later, or by lowering their threshold to respond so that they are exquisitely sensitive at later stages, despite the Notch pathway being dampened in the presence of an inhibitor.”

b) Can the authors show that restricted Notch signalling precedes CS morphogenesis (eg using a Notch reporter transgenic line)? Can timing of Notch be correlated with Hnf1b exclusion from tubule nuclei?

We first tried to use antibodies to NICD but these did not work on zebrafish tissues (at least in our hands). We then examined the Notch reporter (TP1:Venus-PEST) to detect Notch activity in the DE tubule from 18 hpf (Figure 6A and Figure 6—figure supplement 2 in the revised manuscript). We were unable to see any fluorescence in the presumptive CS cells prior to 36 hpf, but did see strong Notch reporter activity in the CS cells from 36 hpf onwards.

In the pronephros, multi-ciliated cells (MCCs) form within the PST and DE tubule segments from a Notch lateral inhibition mechanism between 16 hpf and 24 hpf (as described by Liu et al., 2007; Ma et al., 2007), but we were unable to observe Notch reporter activity in the MCCs prior to 36 hpf. Given this, we speculate that the Notch reporter may not be able to resolve all levels/instances of Notch activity in the pronephros. Later in development (from 36 hpf) the CS cells show very strong Notch activity in the reporter line (we include this new data in Figure 6A of the revised manuscript). To demonstrate the effectiveness of our Notch inhibitor compounds and also the effect of irx3b knockdown on Notch signalling, we have also included the Notch reporter activity in irx3b^gRNA^ and cpdE treated embryos in parallel with our Hnf1b stains in Figure 6B of the revised manuscript. Quantitation of the level of fluorescence in the *TP1:Venus-PEST* embryos has been added as Figure 6D. We discuss the reporter in the revised text as follows:

Results section:

“We then went on to examine a link between Irx3b and Notch, as the Irx factors have been implicated as both positive and negative regulators of Notch signalling ^46,47^. In addition, Notch has recently been connected to CS gland development ^48^ and Notch components are expressed in CS cells (Figure 6—figure supplement 1). We first analysed the *TP1:VenusPEST* Notch reporter line, which contains multiple RBP-Jk-binding sites in front of a minimal promoter ^49^, and found Notch activity in CS cells from 36 hpf onwards (Figure 6A). This result is consistent with the Notch pathway playing a role in CS gland formation. We did not observe fluorescence in presumptive CS cells or tubule multi-ciliated cells (which have previously been shown to form in a Notch-dependent manner ^50,51^) prior to 36 hpf (Figure 6—figure supplement 2), but this may reflect low sensitivity of the reporter.”

and

“Treatment with cpdE also reduced the *TP1:VenusPEST* fluorescence in control (n=28/28) and *irx3b* crispants (n=32/45) consistent with the inhibitory effects of cpdE on CS cell formation being mediated by a block in Notch-Rbpj signal transduction (Figure 6B). Together, these results support a model in which Notch signalling and the cytoplasmic sequestration of Hnf1b in a subset of DE epithelial cells leads to the transdifferentiation of renal epithelial cells into CS cells.

And Discussion section:

“Our data indicate that the transdifferentiation of DE to CS fate occurs in sequential steps and the finding that CS cell numbers are reduced by pharmacological inhibition of Notch signaling from 18 hpf onwards, just prior to the appearance of *stc1*^+^ cells, suggests that the Notch pathway plays an early instructive role. In support of this, we found evidence of Notch-Rbpj signaling in CS cells using the *TP1:Venus-PEST* reporter, although the earliest we could detect fluorescence was 36 hpf, perhaps reflecting the delay needed to accumulate enough VenusPEST protein to be detectable. Interestingly, we also failed to see Venus-PEST fluorescence in developing multi-ciliated cells that are induced by Notch around 16 hpf ^50,51^, therefore the reporter may not be sensitive enough to detect all Notch signaling.“

c) The findings that mindbomb mutants do not generate CS is also opposite to the general findings of Drummond et al. One issue is that by 50hpf, the mindbomb mutant does not form a proper kidney and thus no CS cells can form in the absence of the DE tubules. It appears that the Cadh17 staining is thinner in mindbomb than in WT embryos (Figure 6A). Thus, the statement that Notch is involved in CS formation is misleading and maybe secondary to proper kidney development.

We agree with the reviewer that the mindbomb mutant may have more global effects from the earliest stages of development that may mean any pronephric phenotype at 50 hpf is a secondary effect. Unfortunately, we were unable to characterise Mib mutants in time for resubmission due to the age of our existing fish stocks and failure to identify breeding pairs. Therefore, we have removed this data from the manuscript.

- Please provide further validation and experimental evidence to support conclusions made relating to the role of Notch signalling.2) To claim "transdifferentiation" is somewhat premature. The main concern is that in the strictest sense, transdifferentation occurs when a fully differentiated, mature cell type changes to new cell type and lose all characteristics of the original cell. The main issue here is that the pronephric tubule cells may still be poised in a progenitor state. This study focuses on a process that occurs from 24hpf to 50hpf. This is within larval developmental stages and that many cells are likely to be still poised in a progenitor fate and have not completed differentiation. To sufficiently argue that this is transdifferentiation, one would have to perform a detailed analysis of progenitor markers versus differentiation markers of the kidney tubule.

We have performed further marker analysis in the revised manuscript as additional evidence of the DE-to-CS fate change being an example of transdifferentiation. Specifically, we have examined the expression of the renal progenitor marker *lhx1a* and shown that it downregulates around the time that the intermediate mesoderm transitions to an epithelial tubule (between 14-16 hpf). As this spatiotemporal pattern has already been reported for *lhx1a* (as well as other progenitor genes like Pax2a and Pax8), we have included these data in Figure 2—figure supplement 1. As evidence that the tubule at 22 hpf (just prior to the appearance of CS cells) is in a mature epithelial state rather than being progenitor-like, we have added stains for F-actin (apical tight junction), Collagen IV and Laminin (basement membrane), *slc12a1* (the solute transporter that characterises the functionality of the DE segment), *atp1a1a.4* (a subunit of the Na/K ATPase that is a characteristic of transporting epithelia) and *krt18* (a tough structural protein of epithelia). Together, we believe these markers demonstrate that the tubule at 22-24 hpf does not represent a progenitor population. In the Results section of the revised manuscript we describe these results as follows:

“One possibility is that the renal epithelial cells in the tubular region that gives rise to the CS have retained some progenitor-like state and direct transdifferentiation is not occurring in the strictest sense. To explore this, we closely examined the expression of the renal progenitor marker *lhx1a* from 14-50 hpf. At 14 hpf, transcripts for *lhx1a* are found throughout the intermediate mesoderm in mesenchymal cells but are rapidly downregulated in all tubule precursors by 16 hpf (Figure 2—figure supplement 1), when the transition to an epithelial tube occurs ^35,36^. This transition corresponds with the restriction of other markers of renal progenitor state, such as pax2a^18^ and pax8^37^, although expression of these genes are retained in the DL segment where they likely play later roles in growth ^38,39^. To demonstrate that the tubule has achieved an advanced state of epithelial differentiation prior to *stc1* transcripts appearing, we confirmed at 22 hpf that the DE segment contains apically localised F-actin and basally deposited Laminin and Collagen type IV, indicative of apical tight junctions and a basolateral basement membrane, respectively (added as Figure 2A). In addition, the DE segment at this time point uniformly expresses *slc12a1, atp1a1a.4* and the *krt18* (but not *stc1*), consistent with having the identity of a differentiated transporting epithelium (Figure 2A). At 50 hpf, the DE segment continues to be marked by the expression of *slc12a1, atp1a1a.4* and *krt18* whereas the extruded CS cells do not express these genes. Instead, staining with Phalloidin and antibodies to Laminin and Collagen type IV shows that the CS gland has lost apically localised F-actin and is enveloped by a separate basement membrane (Figure 2A). Taken together, these results support the notion that by 22 hpf, the DE segment has differentiated into a renal tubular epithelium, and that a subset of these cells then transdifferentiate into CS cells.”

- Please provide further evidence that this is indeed a transdifferentiation event by using further marker analysis.3) At many points there is limited quantitative information for each experiment. For example;a) Figure 1 and Figure 2 show individual representative images of IF and ISH stains to show how CS morphogenesis occurs. No n values, measurements or quantitative approaches are used to assign these behaviours and so it is difficult to know how robust these observations are. The authors should provide some form of quantitative analysis of the process.

To quantify CS extrusion, we have used the *cdh17:egfp* line to describe the initial bulging and extrusion of the gland from the tubule at 24 hpf, 32 hpf, 40 hpf, 50 hpf, 60 hpf and 72 hpf. These results are included in Figure 1C of the revised manuscript.

b) Cdh1 staining in Figure 2A is stated to be largely lost at 50hpf. This is not clear in the image shown. Some quantification or at least demarcate the region where this loss of expression is seen should be included. Also, the authors state that F-actin is lost in CS cells at 50 hpf, but there is only one image at 50hpf and not earlier stages to compare with.

We have highlighted the region where there is loss of Cdh1 and described this in the legend, this change can be seen in Figure 1C of the revised manuscript (boxed region at the interface between the CS gland and the tubule). In response to point 2 we have added wholemount lateral views of apically localised F-actin in the DE tubule at 22 hpf and the loss of this F-actin at 50 hpf (Figure 2A). Of note, we also observed Laminin and collagen IV around the CS gland at 50 hpf, which is further evidence of a loss of cell-cell adhesions and replacement for cell-matrix adhesions.

c) Throughout the manuscript n-values are only given for treatments and not for controls. Control numbers should also be provided and statistical analysis provided for all the main findings.

We have now quantitated many aspects of the phenotypes with statistical analysis and p values. For instance, we have added the proportions of embryos that have undergone CS extrusion in Figure 1, quantitated the level of Hnf1b in the nucleus in Figure 3 and Figure 6, measured tubule width and length in Figure 5, and fluorescence intensity in Figure 6. For the more qualitative in situs, we have added in the control numbers (although in all cases these do not show a phenotype, thus have a ‘zero’ proportion). *eLife* also requires a Dryad spreadsheet of all our empirical data and this contains all the empirical results for the experiments shown in the paper.

d) How many mindbomb embryos were counted and were stc1+ cells counted or is the analysis purely qualitative based on the size of the stc1 stained region?

No longer applicable as we have removed the Mib data.

e) The Irx3b mutants – no n-values are provided and only cdh17 expression is analysed. Please provide full details of analysis.

In the revised manuscript we provide n values for the number of embryos that phenocopy the *irx3b* morphants and crispants in progeny from an irx3b^+/-^ in-cross. We also show the phenotypic effects in homozygote mutants on *stc1, slc12a1* and *cdh17* expression analysed by *in situ* hybridisation (Figure 6—figure supplement 4). In addition, we have added the genotype of the stable *irx3b* mutants (a 29 base pair deletion that causes a frame-shift mutation).

f) The authors mostly used non-quantitative approaches to assess the morphological phenotypes in Figure 4 and Figure 5.

In the revised manuscript we include Table 1 (A-C) that shows empirical data (taken from the Dryad spreadsheet) for the number of embryos that had ectopic *stc1* expression and reduced *slc12a1* expression in *irx3b* morphants shown in Figure 4. See comments above in c for a list of quantitative measurements now added.

- Please improve the use of quantification of data and statistics for these points and throughout.4) The concept that Hnf1b is being sequestrated into the cytoplasm in CS forming cells is intriguing as this is one example of how cell fates can change. However, the data shown with antibody staining is not convincing as shown in Figure 3A, Figure 6C and in Figure 3—figure supplement 2.There seems to be some inconsistency in the staining pattern of Hnf1b. The speckled pattern is apparent in the right panel of Figure 3A (50 hpf) and Figure 3—figure supplement 2, but not in the middle panel of Figure 3A (22 hpf) and Figure 6C, in which Hnf1b signals appear to be undetectable in CS cells.In Figure 3—figure supplement 1, hnf1ba mRNA expression was not detected in CS cells. Does this suggest that Hnf1b pathway is inhibited in the transdifferentiated CS cells by transcriptional suppression rather than by cytoplasmic sequestration?- Please improve clear quantification of the staining for Hnf1b and clarify if loss of expression or cytoplasmic sequestration is the major mechanism here.

We have enhanced the contrast and increased the magnification of Figure 3A to more clearly show the cytoplasmic localisation of Hnf1b in the middle panel (note the arrow is pointing to a CS cell and not the speckles). We have added ‘corrected total nuclear fluorescent’ (CTNF) measurements as quantification of Hnf1b in the nucleus at 22 hpf (Figure 3A).

We have replaced the data from Figure 6 with much clearer cryosections and the cytoplasmic localisation of Hnf1b is now more obvious. Quantification of these embryos was also performed, nuclear staining for Hnf1b in the tubule/CS at 50 hpf was measured by CTNF and are presented in a histogram in Figure 6C.